

# Simultaneous, vision-based fish instance segmentation, species classification and size regression

Pau Climent-Perez, Alejandro Galán-Cuenca, Nahuel E. Garcia-d'Urso, Marcelo Saval-Calvo, Jorge Azorin-Lopez and Andres Fuster-Guillo

Department of Computer Technology, University of Alicante, San Vicente del Raspeig, Spain

## ABSTRACT

Overexploitation of fisheries is a worldwide problem, which is leading to a large loss of diversity, and affects human communities indirectly through the loss of traditional jobs, cultural heritage, *etc*. To address this issue, governments have started accumulating data on fishing activities, to determine biomass extraction rates, and fisheries status. However, these data are often estimated from small samplings, which can lead to partially inaccurate assessments. Fishing can also benefit of the digitization process that many industries are undergoing. Wholesale fish markets, where vessels disembark, can be the point of contact to retrieve valuable information on biomass extraction rates, and can do so automatically. Fine-grained knowledge about the fish species, quantities, sizes, *etc*. that are caught can be therefore very valuable to all stakeholders, and particularly decision-makers regarding fisheries conservation, sustainable, and long-term exploitation. In this regard, this article presents a full workflow for fish instance segmentation, species classification, and size estimation from uncalibrated images of fish trays at the fish market, in order to automate information extraction that can be helpful in such scenarios. Our results on fish instance segmentation and species classification show an overall mean average precision (mAP) at 50% intersection-over-union (IoU) of 70.42%, while fish size estimation shows a mean average error (MAE) of only 1.27 cm.

# INTRODUCTION

The overexploitation of fisheries is a problem that affects most seas in the world. Many stakeholders are involved in the fishing industry, each with different interests that need to be preserved: from long-term, sustainable exploitation, to the preservation of marine ecosystems for generations to come. However, management of fisheries is a complex task as reviewed by *Gladju, Kamalam & Kanagaraj (2022)*, which currently involves interpolation of statistical data obtained from a small percentage of samples, given the impossibility to sample and process the large amount of incoming catches per day. Knowledge about these catches is necessary for a better assessment of the health of fisheries. Fine-grained and frequent sampling of such data is important, according to *Palmer et al. (2022)*.

Corresponding author
Pau Climent-Perez,
pau.climent@ua.es

This article is framed by the multi-disciplinary project (*DeepFish-Project, 2023*) about fisheries processes automation, focused on providing a system to control the different stages in the fish market. In fisheries, the control of how many species, instances of each specimen, and size of them are critical aspects for legal and business control. Capturing small fishes as well as fishing certain species in restricted periods of the year might break the law. Counting and sizing the specimens can help control the actual catching of the day. Furthermore, estimation of the biomass is derived from the fish size, so it can also be automated after fish size is obtained. As part of this project, in particular, this article aims to segment, classify, and regress fish sizes in wholesale fish markets using machine learning and computer vision techniques.

The Food and Agriculture Organization (FAO) of the United Nations (UN), estimates that small-scale fishing boats represent 80% of the fleet in the Mediterranean (*FAO, 2020*). In their Plan for Action for Small-scale Fisheries (RPOA-SSF) they call for improving the knowledge retrieval on catches, as well as on fisheries status and health. Because of the size of such fisheries, and the direct involvement of all stakeholders, *d'Armengol et al. (2018)* emphasize the importance of shared management strategies, as these increase acceptance by fishers.

Traditionally, small-scale wholesale fish markets often receive the fish caught by these small-scale fishing boats. In these settings, it is not common to have automated, digitized systems for catch counting, fish sizing, *etc*. The quality of this information is, hence, conditioned by a series of cascading, accumulated errors that range from the fishing boat, to the staff on the wholesale fish market, auction, government inspectors, and so on. Given the large amount of fish disembarked, it is often not possible to sample for inspection but a small fraction of all catches of the day. Furthermore, human miscommunication, specially when manually communicating data of fish captures, can lead to increased error rates, and lead to imprecise models.

Solutions based on the use of computer vision might aid this situation, by helping reduce errors caused by the accumulation of human errors. However, their usage is not extended in traditional industries such as fishing. The next section will look at the solutions that have been envisaged so far, and how these can help shape a solution that is aimed at the goal of this article, which is to help in the effort of fisheries health assessment by means of capturing as much information as possible from pictures of fish trays in small-scale, wholesale fish markets. The focus is brought to the classification of fish species in the batches being processed, as well as the estimation of specimen size. This information can be useful to perform further analytics on the data by various stakeholders. An example of this would be estimation of biomass extraction rates from species and fish size information, to be performed by marine biologists.

## PREVIOUS WORK

The review by *Gladju, Kamalam & Kanagaraj (2022)* compiles different types of applications of data mining and machine learning in aquaculture and capture fisheries. Applications in aquaculture include monitoring and control of the rearing environment, feed optimization and fish stock assessment. As an example, widespread applications in

aquaculture are fish counting, fish measurement and behaviour analysis (*Yang et al., 2021*; *Zhao et al., 2021*; *Li, Hao & Duan, 2020*). Similarly, applications in fisheries comprise resource assessment and management, fishing and fish catch monitoring and environment monitoring (*Gladju, Kamalam & Kanagaraj, 2022*). In recent years, due to the digitization efforts by governments, including public funding aimed at this direction for industries, a number of examples of fish market and fishery management systems, and digitization projects have appeared. Some of these are focused on management, for instance the studies of *Bradley et al. (2019)* and *Clavelle et al. (2019)*. The use of deep learning techniques for fish detection and measurement is more recent but rapidly increasing. *Giordano, Palazzo & Spampinato (2016)* focuses on fish behaviour analysis from underwater videos. *Marrable et al. (2023)* proposes a semi-automated method for measuring the length of fish using deep learning with near-human accuracy from stereo underwater video systems. *Álvarez-Ellacuría et al. (2020)* propose the use of a deep convolutional network (Mask R-CNN) for unsupervised length estimation from images of European hake boxes collected at the fish market. *Vilas et al. (2020)* address the problem of fish catch quantification on vessels using computer vision, and *French et al. (2019)* the automated monitoring of fishing discards. However, none of the reviewed works above focuses on the analysis of images with varied fish species on auction trays at the fish market.

Since this article focuses on the problem of automatic fish instance segmentation (IS), including species identification, and size estimation, an analysis of such specific, previous works is deemed necessary.

In computer vision, image classification is a family of methodologies which attempt to determine the class of an image (*e.g.*, dog, cat, chair, table, *etc.*), from a series of pre-defined classes (labels). This can be done either using the whole image as input to the method, or using parts or regions of interest of the image, that might have been extracted from an object detector. This field has been vastly studied, but is still of relevance in current computer vision research efforts. So far, the best results have been achieved *via* deep learning, that is, using neural networks for classification such as the cases of *Zhao et al. (2017)* or *Minaee et al. (2021)*. Image segmentation, on the other hand, is a field of computer vision that comprises methods that can label images at the pixel level, thus generating masks with the same value for all pixels belonging to a certain class of objects, or textures; and different colours are used to label different classes of objects and textures (semantic segmentation). However, when combined with object detection (that usually provides a bounding box as an output), and each detected object is given a different identifier, one talks about IS. For instance, in this article, each fish in the tray is given a different identifier, even in the case in which several of the fish shown are of the same species (class). The review of *Garcia-Garcia et al. (2018)* and *Hafiz & Bhat (2020)* provide an in-depth study on this topic. Furthermore, image segmentation for fish classification has been studied in several articles. *Rauf et al. (2019)* use a modification of the VGGNet, whereas *Zhang et al. (2020)* proposed an CNN-based architecture for automatic fish counting; finally, *Hasija, Buragohain & Indu (2017)* use Graph-Embedding Discriminant Analysis for robust underwater fish species classification, yet it does not provide real-time classification capabilities, which limits its application for fast-paced environments. In

contrast to that, YOLO ('you only look once') proposed by *Redmon et al. (2016)*, is an object detection network known for its simplicity and efficiency (with real-time capabilities). It has been used in underwater object detection by *Sung, Yu & Girdhar (2017)* and *Zhang et al. (2021)*. The latter presents a model composed by MobileNet v2, YOLO v4 and attention features for fish detection. A more recent work by *Marrable et al. (2022)* use a later version of the network, YOLO v5, for fish detection and species recognition. *Pedersen et al. (2019)* developed a fish dataset and use YOLO v2 and v3 as a baseline for evaluation. There exist other alternatives of instance segmentation based on deep learning architectures, such as Mask-RCNN (*He et al., 2017*), RetinaMask (*Fu, Shvets & Berg, 2019*), or FCIS (*Li et al., 2017*).

In spite of the existence of several methods, the YOLO model has outperformed previous networks for object detection in terms of speed, and has raised interest in the object detection community, as proven by the many variants that have been published since it first appeared. This fact, combined with the need for instance segmentation (*i.e.*, the provision of masks), and not just object detection (*i.e.*, object bounding boxes), has led to the creation of YOLACT by *Bolya et al. (2019)*. In their approach, which stands for 'You only look at coefficients' they use a two-stage architecture: first, prototype masks are generated (in the *Protonet* subnet); later, a set of coefficients is predicted per detected instance. Furthermore, a later proposal termed YOLACT++, by *Bolya et al. (2022)*, improves the segmentation by means of several improvements, namely: adding a fast mask re-scoring branch, which improves the correlation between the mask generation and the class confidence; as well as by adding deformable convolutions in the backbone; and a faster version for the non-maxima suppression (fast NMS).

This article proposes an architecture for segmenting and measuring fish specimens in fish trays, by using YOLACT network and a size regressor in a combined manner, as it is explained in detail in "Proposal".

# PROPOSAL

The main contribution of this article is a system to automatize the processes of fish instance segmentation (IS) and size regression. As part of larger research project (*DeepFish-Project, 2023*) this contribution is embedded in an edge-cloud based system for fish markets. The edge-cloud paradigm brings part of the processing to the end nodes, that is, to decentralise the computation.

In particular, this project aims to segment, classify, and regress fish sizes in wholesale fish markets. In order to do it, images are obtained from a standard camera (see Experimentation and Results) and passed to a network architecture that performs IS and fish species classification, coupled to a fish size regressor (See Fig. 1).

A YOLACT network is trained for the IS task, and its outputs are used for the regression of fish sizes. To train the IS network, human labelling is provided for all uncalibrated images of fish trays shown during training. Furthermore, this human labelling provides information regarding tray corners (specifically tray handle corners, in this case) in order to make it possible to calculate the ground truth fish sizes using visual metrology to estimate a correspondence (homography) between the points of the corners of the tray on

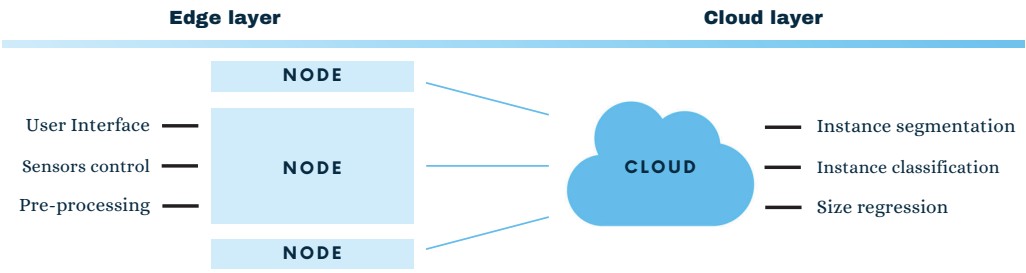

**Figure 1 Edge-cloud architecture for smart fish market systems.**

the image, and the plane represented by the actual corners of the tray in the real world. Knowing the size of the tray in the real world, and *via* the estimated correspondence, it is possible to estimate the sizes of the fish specimens present on the tray, given that the correspondence can be used to transform the size of any area in pixels representing a fish on the tray to centimetres. This process of corner-labelling and homography estimation for each image, however, is labour-intensive and therefore is only provided for training images. The regressor module of the proposed approach is therefore required to learn the conversion internally, and to estimate fish sizes from uncalibrated images directly (from a similar angle of incidence and distance). This is because smaller-scale fish markets, as noted, might not have the budget or required facilities for a fixed camera installation which is typically mounted overlooking an automated conveyor belt, and therefore, images may be taken using portable electronic devices with an embedded camera (smartphones, work tablets, *etc.*).

The information extracted by the proposed system is aimed at fish stock managers, which can gather relevant information about the health status of exploited stocks, derive biomass extraction rates, *etc.* This is not only useful to managers but to all stakeholders involved (*e.g.*, fishers, consumers, local governments, *etc.*), since it can help take informed decisions based on accurate evidence, including information on fish species caught, the sizes of specimens per species, the total biomass of said specimens (which can be derived from their size, or from other visual cues), *etc.*

To address the problem of training the IS neural network in the main contribution, a second contribution of this article consists in the gathering and preparation of a large dataset of fish trays from local wholesale fish markets. This is the *DeepFish* dataset. It consists of 1,100 images of fish trays from the small-scale wholesale fish market in El Campello, and contains more than 7,600 fish exemplars in total. The images were taken from March to October 2021, with a majority of images taken in the first three months. Further details about the process and the resulting dataset can be found in *García-d'Urso et al. (2022)*. Furthermore, the dataset is available online for download from a public repository by *Fuster-Guilló et al. (2022a)*.

The general overview of the proposed methodology is shown in Fig. 2, which consists of two main workflows. In the top, with a blue background, the workflow for training the IS network (using YOLACT), as well as the regressor for fish size estimation, is shown. The

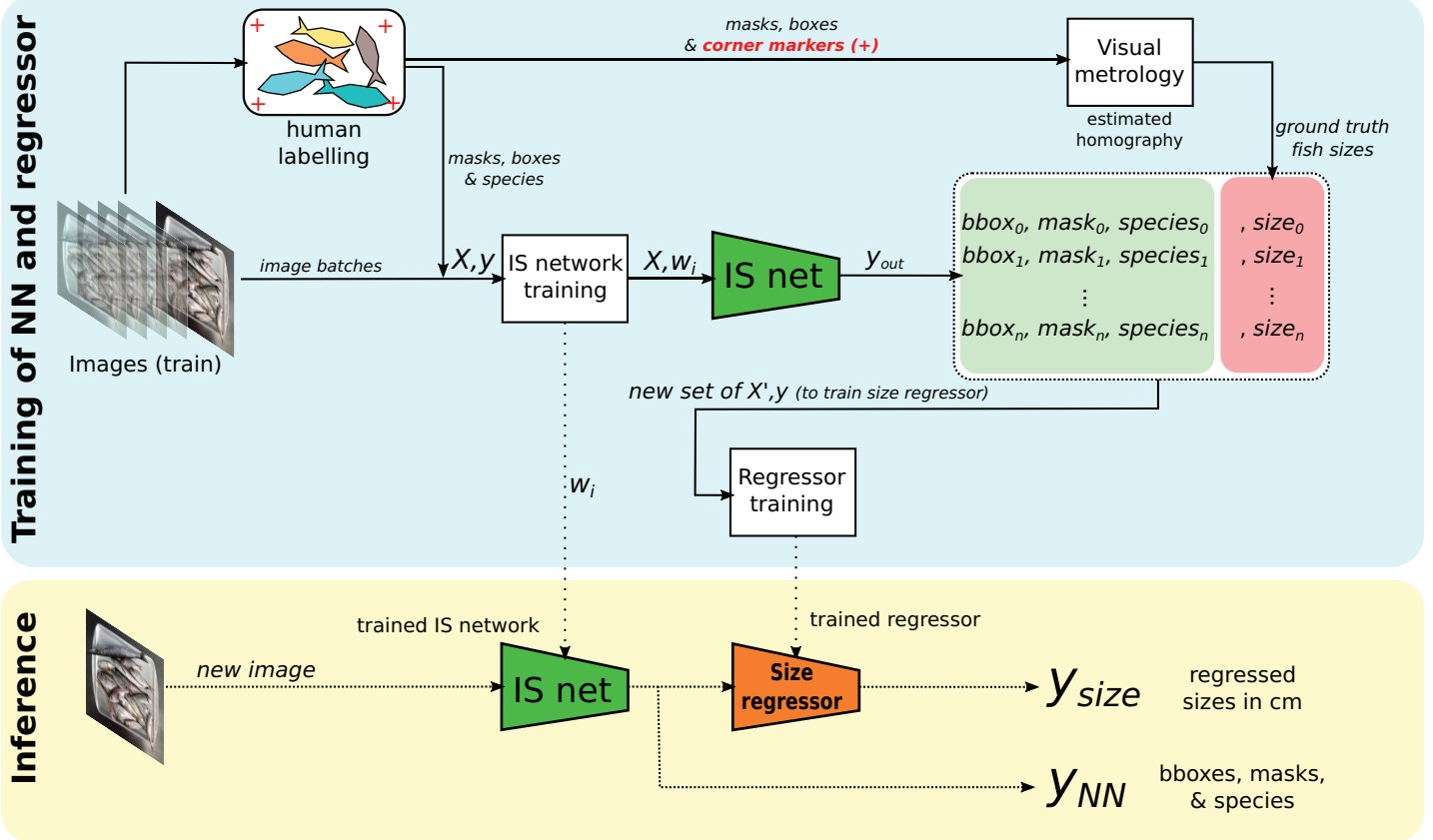

**Figure 2 Overview of the proposed method for instance segmentation (IS) and size regression.** At the top, in a blue box, the training process. At the bottom, in the yellow box, the inference process for new images.

bottom part (in yellow) shows the workflow for new images once the system has been trained.

Next, in "Method", the different components that make up the proposed system are presented. The experimental setup and results follow (see "Results and Discussion" to better assess the performance of different variants of YOLACT for the IS neural network, a comparison of different backbones (ResNet of different sizes), as well as different YOLACT variants (*i.e.*, original *vs* YOLACT++) is included. Similarly, results with different regressors will be compared, showing the results-driven approach taken to select the best-performing regressor for the final system. Then, results for the overall system are presented. Finally, some conclusions will be drawn, and work left for the future, outlined (Conclusions).

## METHOD

The entire proposal is composed by different elements, from one side the edge layer with all the user interface parts and the data pre-processing, to the cloud layer performing the heavy computation. Since the main computational burden of the proposed system is carried by the cloud side, this section will introduce in detail the learning architecture.

Later, a description of the specific needs for this project in the edge layer are presented in "Edge Layer Computing".

The cloud layer is made up of different components (Fig. 2) that work in conjunction to provide two outputs at the end: $y_{size}$ for the estimated fish size, as well as $y_{NN}$ which contains the information of bounding box, mask, and species label for each fish segmented from the image by the IS neural network. To do this, two main modules are required: the IS network, and the fish size regressor. Each of these will be introduced next.

## Instance segmentation and species classification

The function of this component in the system (the IS network) is to perform instance segmentation of fish specimens present in the trays and to be able to classify said specimens according to their species. Instance segmentation, as said, is different from object detection in that the output consists of a mask (including a class label, and identifier) per specimen, and not just a bounding box per detected object. Furthermore, instance segmentation differs from 'classical' segmentation in that it does not provide a single label for all areas of the image that pertain to the same class, but it provides separate masks (with different identifiers) for detected objects even when these have some overlap in the image (*i.e.*, different from *semantic* segmentation). Several options would exist for this module, as it was mentioned in "Previous Work", however, YOLACT is chosen due to its real-time capabilities, and its comparative results in terms of mean average precision scores (mAP scores) for the MS COCO dataset as it is presented in the original article by *Bolya et al. (2019)*.

Because this module is based on a neural network, which falls under the umbrella of data-driven methodologies, a step of paramount importance is the collection of relevant data (*i.e.*, data exemplars for the problem at hand). Furthermore, preprocessing, and augmentation, will also need to take place. Preprocessing in this context refers to adapting the data to the network input format, for instance: resizing images to $550 \times 550$, normalizing the RGB color data from [0..255] to [0..1], *etc*. Data augmentation is explained later in detail in "Augmentation". This data collection is important for systems, like the proposed one, in which transfer learning is to be carried out, since the new data ought to modify the weights on a small scale as to enrich the network, *i.e.*, improve its recognition capabilities for the new task; but at the same time preserving the original weights in the earlier stages (layers or blocks of them), that are common to different problems. This happens because, usually, networks come pretrained with datasets with millions of images, and the earlier blocks of layers tend to focus on coarser edge and shape features of different areas of the image (*i.e.*, like used to be the case in classical computer vision filters, *e.g.*, Gábor).

As shown later in the Experimentation section, several backbones will be tested, for comparison, *i.e.*, to allow for a performance *vs* model size evaluation. Regardless of the backbone network used, the 'P3' layer of the feature pyramid network (FPN) is connected to 'ProtoNet' which is a fully convolutional neural network in charge of prototype mask proposal. Masks generated this way will have the same size as the input images (*i.e.*, coordinates match). The viability of the generated masks is assessed in parallel, by a
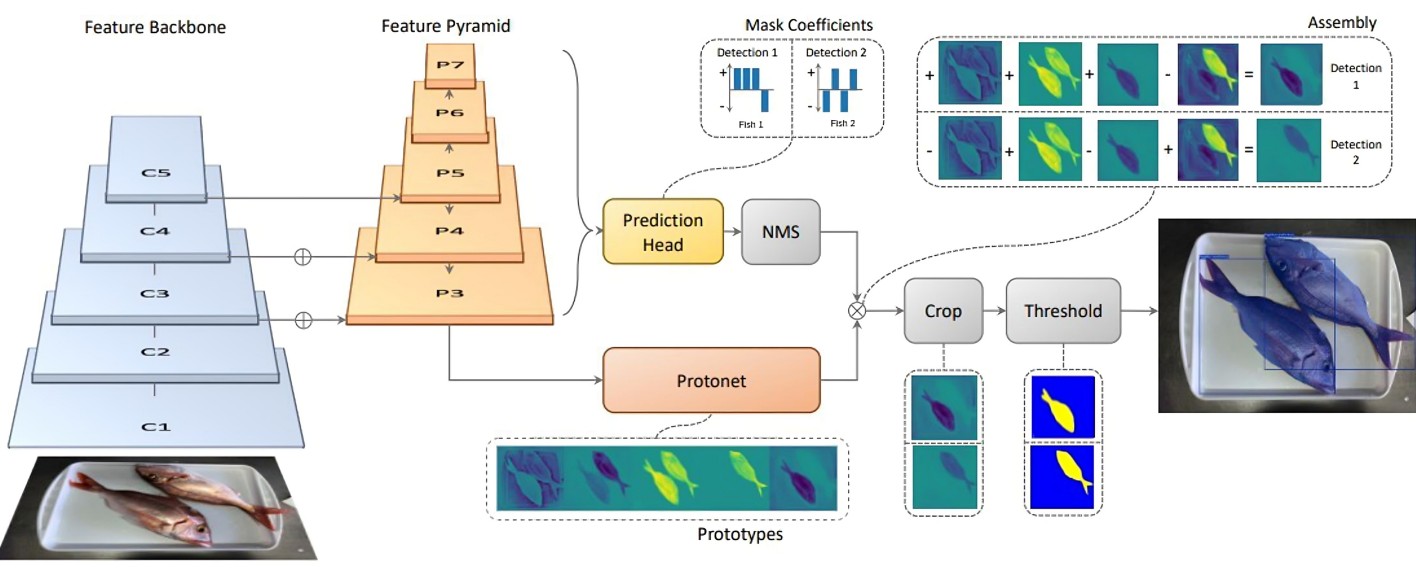

**Figure 3** Diagram of the adapted YOLACT network for fish instance segmentation and species classification (IS), used as part of the proposed approach.

prediction head in charge of finding mask coefficient vectors for each 'anchor' (that is, each layer of the FPN). After masks have been assessed, non-maxima suppression (NMS, or Fast NMS for YOLACT++) is used to discard overlapping mask proposals, and therefore obtaining only one mask per segmented instance. Following that, ProtoNet mask proposals and NMS results are merged. This is done by means of a linear combination, *i.e.*, a matrix multiplication, which is efficient in terms of computational time. Finally, some refinements are applied, consisting on cropping and thresholding, which results in the final mask predictions, bounding boxes, and labels. Figure 3 shows an overview of the adapted YOLACT architecture presented by *Bolya et al. (2019)* for the case of fish instance segmentation and species classification.

## Fish size regression

Estimating the size of exemplars is of great relevance in the field, given that such approaches do not currently exist, and would be very relevant for the stakeholders involved. In our case, from a set of image-derived features rather than the raw images is a novel and robust approach. In this way, our approach decouples the original information from the regressor. Furthermore, this does not only have applications in the field of fish markets or fisheries health assessments, but also in other industrial processes such as fruit size classification, assembly lines processes, *etc*. Following the workflow of Fig. 2, the output of the IS neural network (denoted 'IS net') is a $y_{out}$ which consists of the masks, bounding boxes, and species labels. These then become a new $X'$, that is an input to perform the 'regressor training' (box in the figure), resulting in a trained regressor, denoted

by the orange 'Size regressor' module in the inference part of the figure. To learn the sizes, a ground truth $y_{gt}$ is required.

It is also important to highlight that the reliability of the results depend on taking the images roughly at the same distance from the trays. In the fisheries scenarios the setups do not change over time, however, in a different setup a re-scaling might need to be applied. This $y_{gt}$ is automatically obtained for human-labelled images in the training set, by using points from the tray. Since all trays are of a known shape (rectangular), same size, and have the handles in the same locations, the rectangle formed by the start of these handles is used to obtain the image deformation parameters (in terms of affine transformations). Handles, instead of tray corners, are used because of the particularities of the used trays which happen to have curved corners, which make it difficult to estimate their exact position when labelled by humans. These deformation parameters are then used to obtain a *corrected* image, as well as a *corrected* set of masks and bounding boxes, from which fish sizes can be derived. This process involves the use of 'visual metrology' to estimate the homography between the real-life tray rectangle and the rectangle as observed in the image. The resulting fish sizes are then used as the required $y_{gt}$ in the process of the 'regressor training'. Once the resulting 'size regressor' module is trained, new images can be provided and will result in fish sizes being estimated in an unconstrained fashion, without the need of camera calibration, as long as images are taken from a similar angle of incidence and distance to the fish trays.

To validate this approach, several types of regressors have been used, as will be observed in the Experimentation section below, specifically in "Proposed Size Regression Experiments". The final result of the proposed system is therefore twofold: on the one hand $y_{NN}$ (from "Instance Segmentation and Species Classification" above) will contain information about the masks, bounding boxes, and species labels of fish specimens; whereas on the other hand $y_{size}$ will contain the sizes of said specimens.

## Edge layer computing

On the other side of the edge-cloud presented architecture in Fig. 1, the edge layer unburdens the system by processing part of the information in the end node. For the case of a realistic fish market scenario, most cases will include a friendly user interface to help non-experts in capturing the data, the actual pre-processing and filtering of the data and communication aspects.

Different sensors may collaborate simultaneously, for instance, a code reader for label or tags information acquisition plus a color camera for taking visual images. In our proposal, we use two different cameras for the code (QR code) reading and fish tray images. We propose a color camera to be more adaptable to different codes. In this case, once the code is read and the metadata is stored, the second camera is activated. This is an RGB-D sensor for our particular case, characterized by providing color and distance information simultaneously in a single device. In the case of this article, depth information is not used and hence only color camera might be sufficient, but having this information might help in future works of this project regarding biomass estimation, by including volumetric information.

With the information stored, the system needs to send the data to the cloud layer to perform the more computationally expensive processing. The communication shall be bidirectional to allow not only data transmission but also remote control of the edge node for maintenance or any other purpose. This shall be done using encrypted protocols and, in case the user interface wants to be transmitted, other protocols can be implemented allowing video sequence remote visualization.

## EXPERIMENTATION AND RESULTS

This section will present different batches of experiments that were carried out to validate the presented approach. First, the dataset that was used in the experiments will be introduced. Then, each module, *i.e.*, the 'IS network' and the 'Fish size regressor' will be validated separately, each with a set of experiments aimed at demonstrating the behaviour of the different modules. Finally, an overall validation will be conducted for the whole proposed system.

### Dataset

The current work is part of the DeepFish 2 project (*DeepFish-Project, 2023*), which is aimed at the improvement of fish biomass extraction calculations for different stakeholders, from different data sources. The collaboration with different wholesale fish markets of different scales in the province of Alicante, Spain, has been at the core of the project. The images used in this article correspond to the small-scale wholesale fish market of El Campello, and were captured for six months (May to October) during 2021. The images were captured with a smartphone camera, that was not fixed to any structure, but were all taken from a similar distance and angle of incidence. The images of the market trays include a variety of fish species (see Fig. 4), with a distribution of fish species as depicted in Fig. 5. There are a total of 59 species, of which 18 are considered *target* species due to their commercial value; of these, 12 are kept for the experiments, since a minimum of 100 specimens per species is considered necessary to train the neural network. This number was calculated through experimental validation. These 12 species translate into 13 class labels, due to the sexual dimorphism displayed by *Symphodus tinca* specimens, which are therefore considered under two different class labels. The resulting dataset contains 1,185 images of fish market trays, containing a total of 7,635 fish specimens. Examples of ground truth labelling can be found in Fig. 4.

A modified version of the *Django* labeller by *French, Fisher & Mackiewicz (2021)* is used by expert marine biologists to provide the ground truth for all images in the dataset, including silhouette information, bounding boxes, species label, as well as the size, which is provided as a polyline from the mouth to the base of the tail. Using polylines in fish size measurement is a common practice in this area, as shown in the review by *Hao, Yu & Li (2016)*. Other measurements are also provided, such as the width at the waist, or the eye diameter. This is useful to derive total fish size for partially occluded exemplars, as explained by the consulted experts in marine biology which collaborated in the study. Conversion tables exist in the literature to convert between these alternative measurements and fish size estimates. With this labelling tool, an initial JSON file is generated, which can

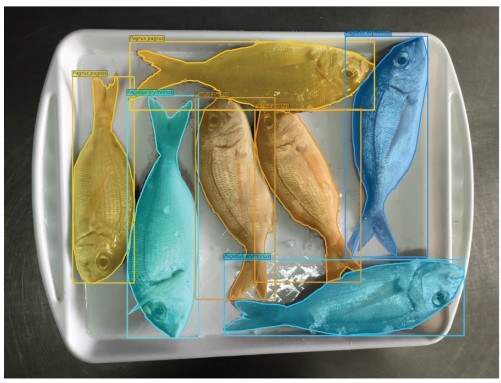
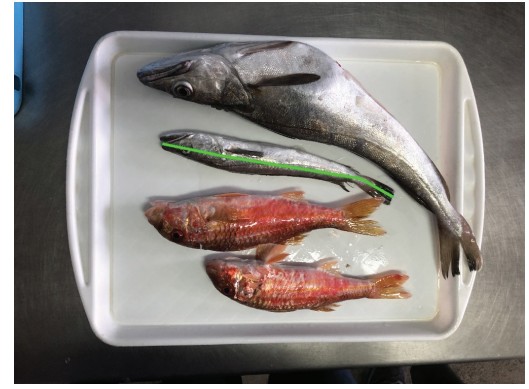

**(a)** Masks, boxes, and species       **(b)** Size ground truth polyline

**Figure 4 Visualization of ground truth data.** For each instance in an image, the human-provided ground truth contains (A) masks, bounding boxes, and species labels (different colours); as well as (B) fish sizes as polylines (one per instance, only one shown).  

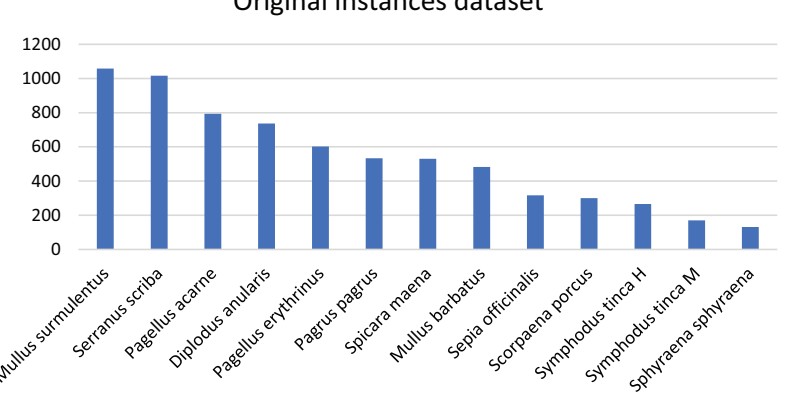

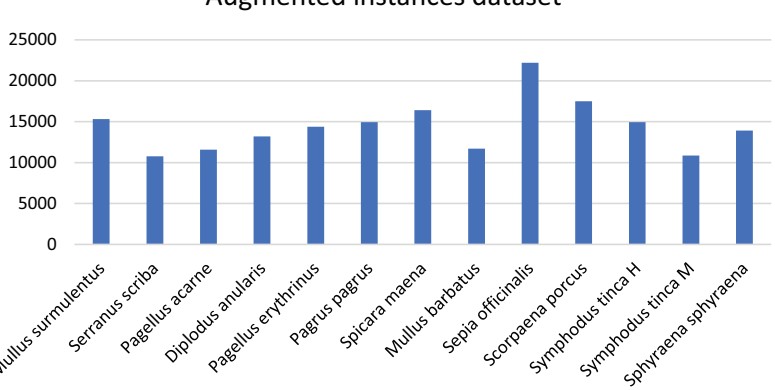

**Figure 5 Distribution of fish species in the DeepFish dataset for the selected species.** The top bar plot presents the original distribution and the bottom bar plot depicts the augmented distribution.

then be converted to an 'MS COCO'-compatible JSON format, *via* a provided script by *Fuster-Guilló et al. (2022b)*. This latter JSON file can then be directly fed to a network for training.

Further details can be found in *García-d'Urso et al. (2022)*. Additionally, the dataset is publicly available for download and described by *Fuster-Guilló et al. (2022a)*.

### *Augmentation*

Since the dataset is highly imbalanced (as made evident from Fig. 5, top plot), data augmentation is used to train the neural network module. After analysing different tools for data augmentation including the proposals of *Buslaev et al. (2020)* and *Jung et al. (2020)*, and considering that it should be able to not just perform augmentation on the data, but also modify the ground truth according to the data transformation applied (*i.e.*, generating a modified ground truth JSON file), CLoDSA from *Casado-García et al. (2019)* is chosen.

Data augmentation is carried out here by applying rotations (15°, 45°, 90°, *etc.*) and translations (5 to 50 pixels) on the images of trays. It is worth mentioning here, trays contain specimens of several species each, and therefore augmentation needs to be carried out taking into account the species that are present in each tray. Yet, a perfect augmentation, in which all species have the exact same number of specimens, is not possible. What is possible, however, is to reduce the difference in specimen numbers after applying the augmentation. This has carefully and manually been done, by augmenting images with the least present species more than those with species for which there is an abundant number of exemplars. Before normalization, the differences between the most common and the least common species is two orders of magnitude ($1 \cdot 10^3$ *vs* $7 \cdot 10^1$), whereas after the augmentation, the number of specimens for all species have the same order of magnitude ($1 \cdot 10^5$ to $2 \cdot 10^5$). The initial number of images of trays is 1,260, of which 1,108 are used for training. Only trays used for training are augmented, yielding a total of 44,366 images in the training set. The new distribution of species after augmentation is shown in Fig. 5, bottom plot. Despite the unequal number of instances per species, after augmentation, the dataset is more balanced. The reader should note that, because of how the specimens of some species are distributed among many trays they appear in a large percentage of the images, and, as a consequence, the augmentation of images will increase those specimens by a larger scale than other species that are not present in as many trays. For instance, *Sphyraena sphyraena* is initially the species with the fewest instances, but it is distributed in many trays along the dataset. After applying data augmentation at the image level, it becomes the most represented species.

This, however, is not the only augmentation applied to the images. Further on-the-fly augmentations are applied during the neural network training process, as part of YOLACT. These consist of: photometric distortion (*i.e.*, altering the hue and saturation), expansion and contraction (*i.e.*, simulating detection at different scales), random sample cropping, as well as random flipping of the images (mirroring).

## Proposed IS experiments

The experiments regarding the IS module are aimed at showing the performance of a set of YOLACT variants, and demonstrate their utility for the task at hand. There is a balance between backbone size, performance, and inference times (which are well known for these variants by *Bolya et al. (2019, 2022)*.

Four different variants are evaluated, by mixing different ResNet backbone sizes (50, 101 or 152 layers), and employing either YOLACT or the improved YOLACT++. The four combinations are: two tests using the original backbone size with either YOLACT/++ variants, as per the original specifications. Two additional tests: increasing the number of layers to 152 for the 'weaker' variant (classical); and decreasing the number of layers for the 'stronger' (++) variant. The rationale behind this is, that this way, the contribution of the backbone size and the variant type can be separated, similar to an ablation test. That is, the classical variant is given a larger backbone to check whether the backbone size alone is capable of compensating '++' variant improvements. Furthermore, the '++' variant is provided with a smaller backbone, to check how much the improvements of that particular variant contribute to the overall results.

In all cases, training parameters stay the same: the input size is $550 \times 550$ pixels, the batch size is of eight samples, training is let to run for 300,000 iterations (62 epochs), with a learning rate (LR) schedule: LR starts at $10^{-4}$, and is further reduced after 200,000 iterations to $10^{-5}$, and then further at 275,000 iterations to $10^{-6}$. Stochastic gradient descent (SGD) is used in all cases as the optimizer, and is configured with a value of $\gamma = 0.1$, with a momentum of 0.9 and decay of $5 \cdot 10^{-4}$.

Regarding the loss function used, it has three components: a classification loss $L_{\text{cls}}$, a box regression loss $L_{\text{box}}$, and a mask loss $L_{\text{mask}}$; with weights of 1.0, 1.5, and 6.125, respectively. Both $L_{\text{cls}}$ and $L_{\text{box}}$ are defined as done in *Liu et al. (2016)*. To compute the mask loss, a pixel-wise binary cross entropy (BCE), Eq. (1), is taken among the set of assembled masks $M$ and the set of ground truth masks $M_{\text{gt}}$:

$$L_{\text{mask}} = BCE(M, M_{\text{gt}}). \tag{1}$$

## Proposed size regression experiments

For the validation of the regression module, several regression models will be compared in terms of accuracy and performance. The regression model employed will be required to perform fish size estimation, and additionally, learn the image calibration required to transform the images during training, given the ground truth fish sizes estimated *via* visual metrology (*i.e.*, the calculated homography). For this part of the system, a series of five experiments is proposed: first, select a subset of best-performing regressors, from the 25 most common in the literature; then, reduce the selection further by checking their performance with hyperparameter tuning; following that, select algorithms that perform the best after normalization of the data; next, apply a 10 $k$-fold validation, and verify the results; and, finally, compare the results obtained to those employing the corner data (*i.e.*, image calibration information). Please note this last experiment consists of providing data,

**Table 1 Mask mean average precision on test dataset.**

| Network | Backbone | $mAP_{50}$ | $mAP_{60}$ | $mAP_{70}$ |
|---|---|---|---|---|
| YOLACT | ResNet-101 | 57.32 | 51.24 | 42.26 |
| YOLACT | ResNet-152 | 65.99 | 60.65 | 48.70 |
| YOLACT++ | ResNet-50 | 68.81 | 66.88 | 60.78 |
| YOLACT++ | ResNet-101 | 70.42 | 68.86 | 62.88 |

*i.e.*, the tray handle corner data, that would not normally be available at system runtime, since it consists of human-labelled data that is provided only during training. However, for the sake of completeness, and to verify the performance of the system in this *ideal* situation, this experiment is also included here.

As will be seen from the initial results, the gradient boost regressor (GBR) model defined by *Zemel & Pitassi (2000)*, extra trees (ET) proposed by *Geurts, Ernst & Wehenkel (2006)*, and categorical gradient boosting regressor (CatBoost) presented by *Prokhorenkova et al. (2018)* seem to be the models with a better fit to the data. This is why these are selected for subsequent experiments. However, for the sake of completeness, support vector machine (SVM) (*Suthaharan, 2016*) variants have been included in all experiments, as a baseline for comparison. These SVM variants are: SVM with a radial kernel (which is appropriate for this type of data), as well as SVM with a linear kernel.

## RESULTS AND DISCUSSION

This section will introduce the results, both from a quantitative and a qualitative point of view, for all experiments presented above for the IS module, and the fish size regressor.

### IS results

As explained, the rationale behind the proposed IS experiments, which entail testing different backbone sizes for different variants of YOLACT, is to be able to determine whether a larger backbone for YOLACT would suffice to counter the improvements introduced by YOLACT++. This section introduces the results for the instance segmentation. Table 1 presents the mean average precision (mAP) values for each backbone size and YOLACT variant, for three different overlap acceptance values (50, 60, 70). Here, overlap is defined as the intersection-over-union (IoU) of predicted and true (expected) mask pixels.

Additionally, Fig. 6 introduces a curve plot, in which performance degradation is tested. That is, the *X* axis shows minimum IoU overlap acceptance tolerances, and the *Y* axis shows the mAP at those points. The figure shows a clear gap between YOLACT++ and YOLACT curves, which is indicative of how improvements introduced in YOLACT++ cannot be mimicked by increasing the backbone size on 'classical' YOLACT. In the case of 'classical' YOLACT, backbone size does seem to matter, as ResNet-101 seems to keep better performance as the minimum overlap acceptance tolerance goes up.

Previous results have focused on detection rates, and detection accuracy of the masks (the 'instance segmentation' part of the network). However, if looking at classification

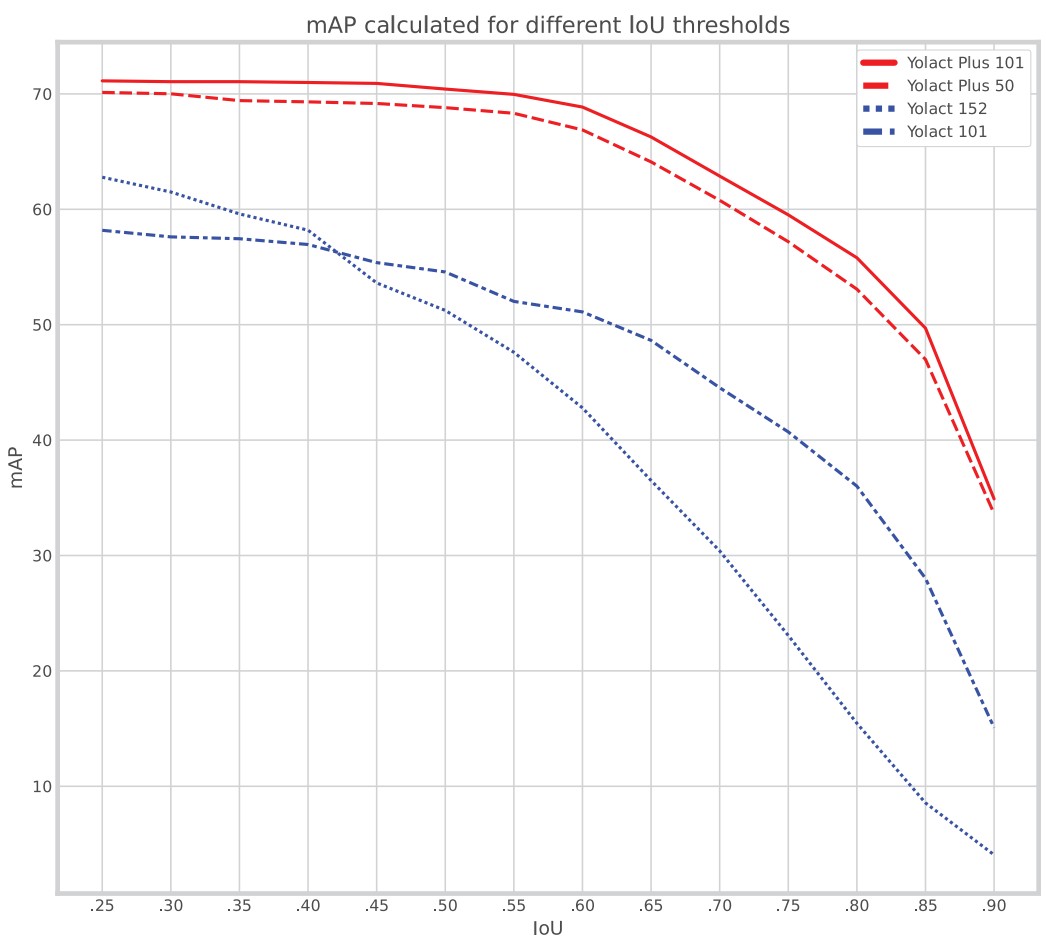

**Figure 6 Mean average precision (mAP) of mask scores at increasing minimum intersection over union (IoU) overlap acceptance levels for all experiments described.**

results per-class (per-species) accuracies, confusion matrices can be plotted. These are shown in Figs. 7 through 10. A particularity of these matrices, is that they all include an additional column (right-most), which accounts for missed detections or false negatives (labelled as 'Missed (FN)'). This value refers to those fish specimens of a specific class label which were manually annotated (*i.e.*, present in the ground truth), but the network detection missed. The color coding of the confusion matrices show darker shade in cell background representing better performance, if it is found in the diagonal of the matrix.

Results of these confusion matrices can be analysed on a case by case basis, leading to some interesting insights. For instance, the first one, for YOLACT with a ResNet-101 backbone, is shown in Fig. 7. The best value in the diagonal can be found for *Sepia officinalis* (91.5%). This will be observed again in the other confusion matrices, and it makes sense, as cuttlefish is the most distinctive species, given it is the only cephalopod in the dataset, and all other classes belong to vertebrate fish species. If looking at other results, it can be observed that males and females of *Symphodus tinca* are slightly confused with

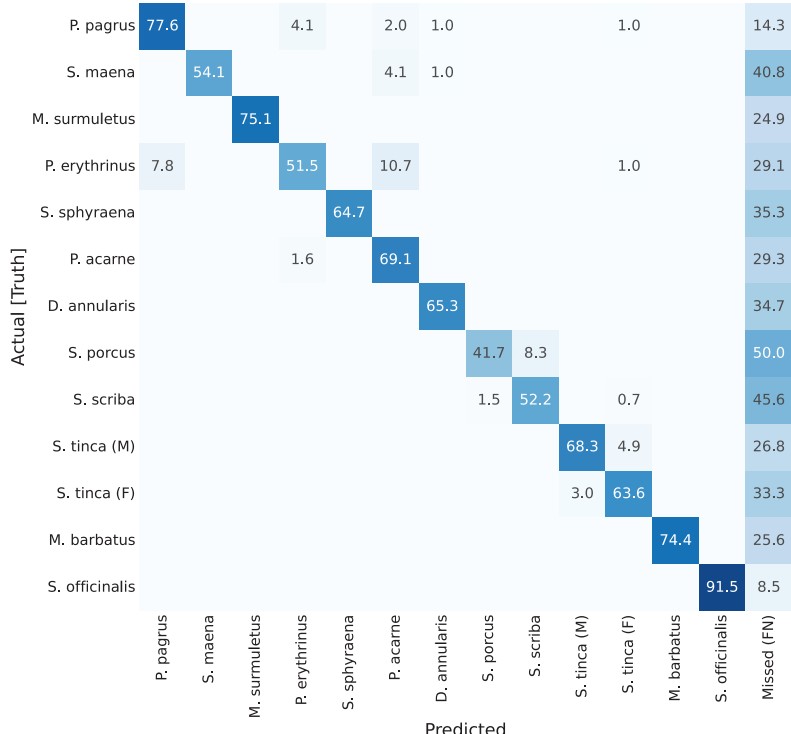

**Figure 7 Confusion matrix for YOLACT network with ResNet-101 backbone.** Values represent percentage (%) of samples, normalized per species (row).

each other (3% and 4.9%). These low values are a result of the common traits of specimens of this species, regardless of its displayed sexual dimorphism. Another observable fact is that, lower values in the diagonal can be attributed to high rates of missed detections, as shown by some darker than usual cells in the right-most column, *e.g.*, *Scorpaena porcus* shows the lowest value (41.7%), with 50% missed detections (FNs), which has a reasonable explanation, as it is the second species with the lowest number of samples, as shown in the species distribution plot in Fig. 5.

Next, on the second confusion matrix (Fig. 8), representing results for YOLACT with a larger backbone (ResNet-152), the best classified species is again *Sepia officinalis*, with 93.6% (as explained). Something else worth mention is the lighter shades in the 'Missed (FN)' column, which shows a general improvement in detection. This was also reflected in Table 1, in which the $mAP_{50}$ value is improved from 57.32% to 65.99% (9% difference). Even *Scorpaena porcus*, the least correctly classified species, shows an improvement in detection, as missed detections drop from 50% to 41.2%. These results indicate that a larger backbone size is beneficial, in this case.

The next two confusion matrices show the results for the YOLACT++ variant. The third confusion matrix, presented in Fig. 9, corresponds to YOLACT++ with a ResNet-50 backbone. In this case, the values at the diagonal are higher for 61% of the cases (species), as indicated by darker shades. This better performance is present even with a smaller backbone size, and is also visible through the mAP values shown in Table 1, as there are

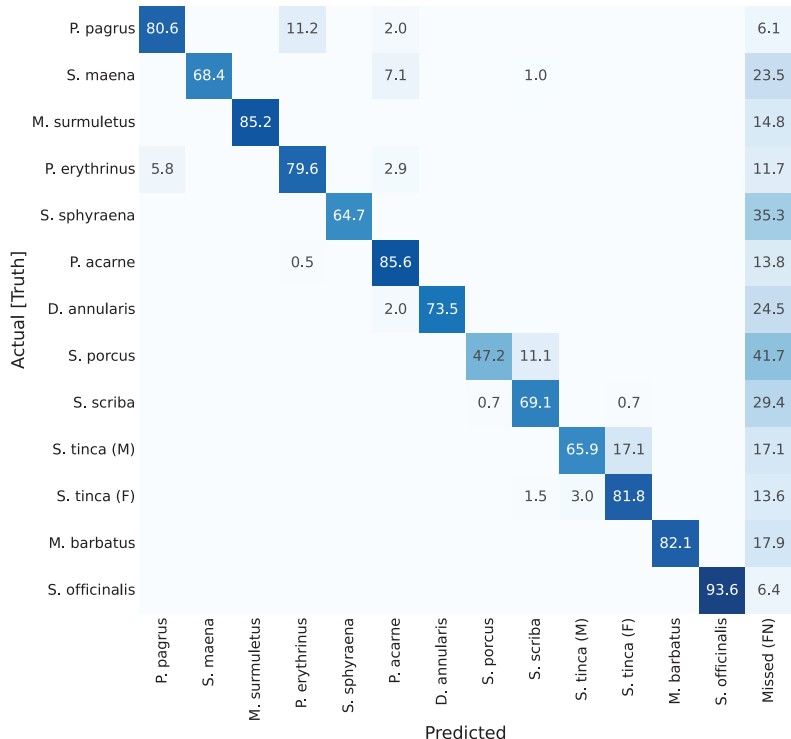

**Figure 8  Confusion matrix for YOLACT network with ResNet-152 backbone.** Values represent percentage (%) of samples, normalized per species (row).

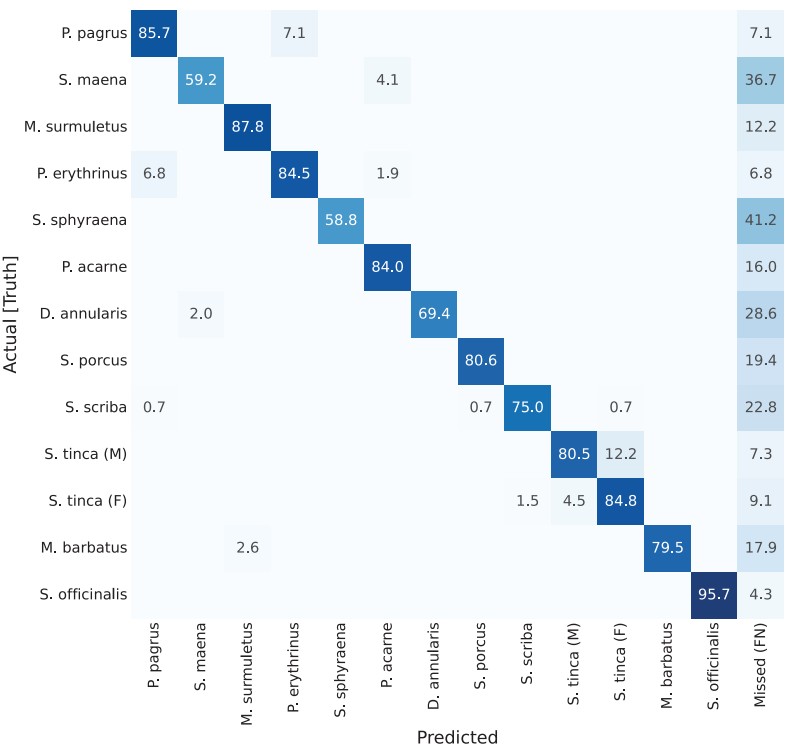

**Figure 9  Confusion matrix for YOLACT++ network with ResNet-50 backbone.** Values represent percentage (%) of samples, normalized per species (row).

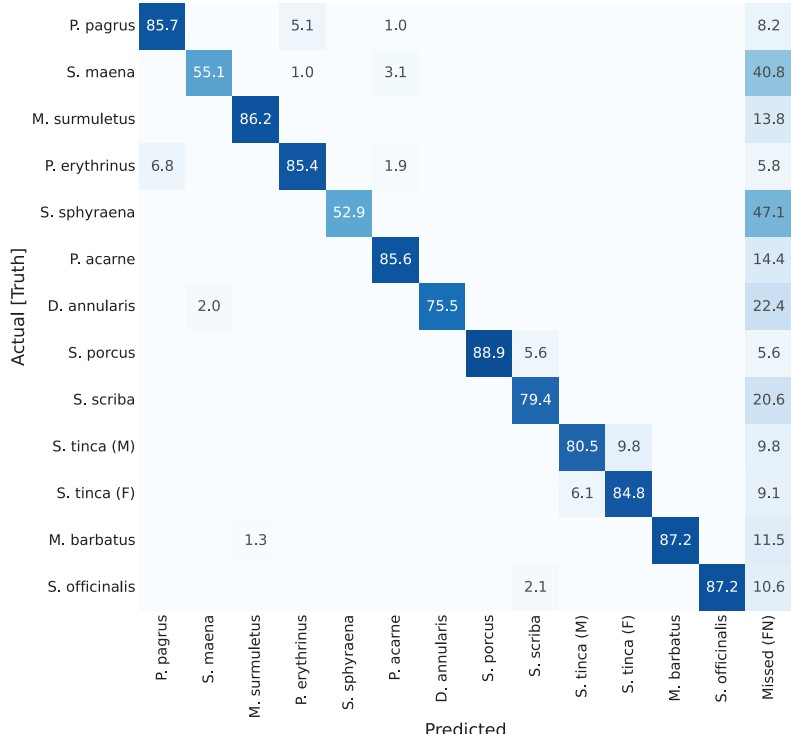

**Figure 10 Confusion matrix for YOLACT++ network with ResNet-101 backbone.** Values represent percentage (%) of samples, normalized per species (row).

3%, 6%, and 12% improvements for the mAP values at 50%, 60%, and 70% minimum overlap requirement, respectively. Note well that this 12% is the highest improvement shown in the experiments. Cuttlefish (*Sepia officinalis*) is still the best-classified species, at 95.7%, which is the highest value for the species so far. All values seem to have increased, as demonstrated by harder examples such as *Scorpaena porcus*, with values around 80% to 85%. The worst score is assigned to *Sphyraena sphyraena* (58.8%), this has several possible causes: a high rate of missed detections, at 41.2%, which can be explained by the low number of specimens registered, and the odd shape of this specific fish species which is very long and can be presented rolled in different ways on the trays (therefore a detection problem, rather than a misclassification problem). However, missed detections (*i.e.*, false negatives) are much lower for all other species. It can be concluded that YOLACT++ improvements can compensate the use of a smaller backbone. This has two additional benefits: first, smaller backbones can usually be trained in less time; and furthermore, a smaller footprint network can be embedded in edge computing hardware platforms, in case it was deemed necessary.

The last confusion matrix corresponds to YOLACT++ with a ResNet-101 backbone (Fig. 10). Contrary to previous tests, *Sepia officinalis* does not show the best results, but other species show improved classification scores, leading to improved overall performance, as shown in Table 1 with mAP scores approximately 2% higher for this test. Specimens of *Scorpaena porcus*, which obtained low classification scores in the 'classical'

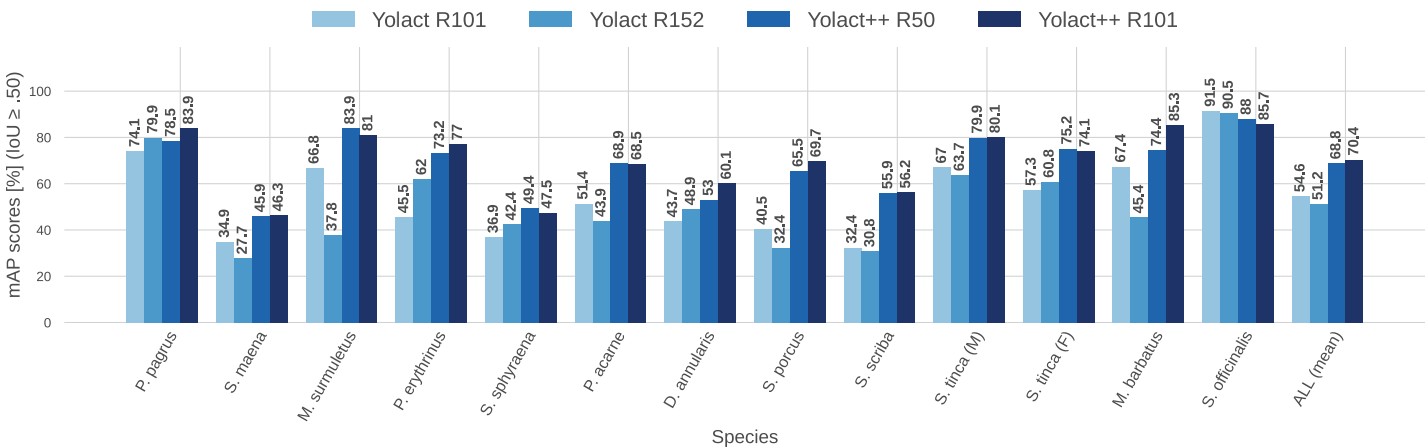

**Figure 11 Per-class (per-species) average precision (AP, in %) for all IS module configurations tested (IoU ≥ 0.5).**

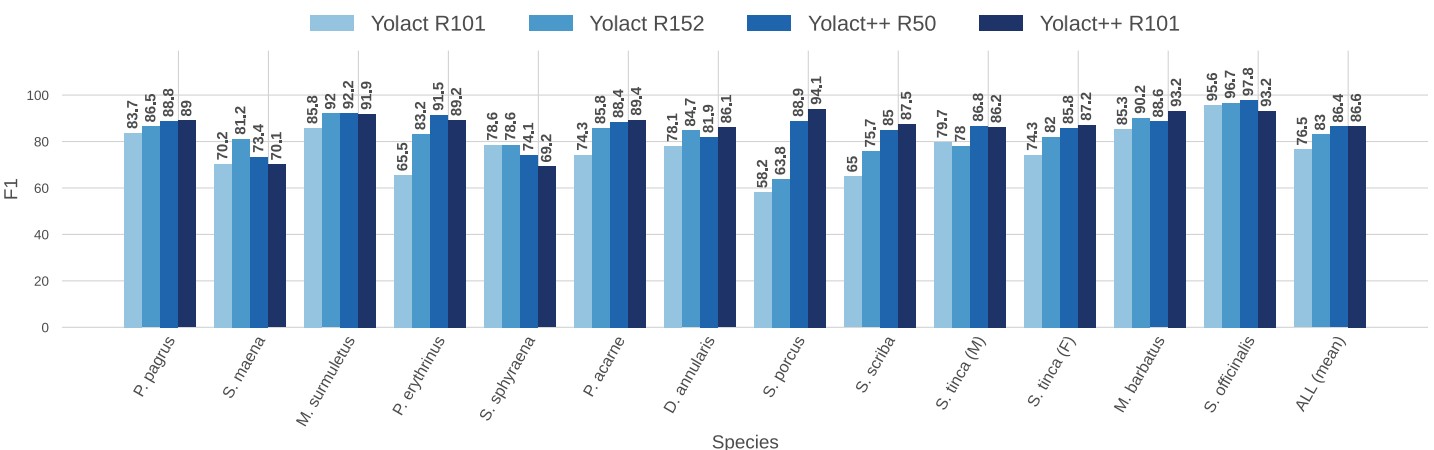

**Figure 12 Per-class (per-species) average F1 score (F1, in %) for all IS module configurations tested (IoU ≥ 0.5).**

YOLACT settings, now show 88.9% scores. However, *Sphyraena sphyraena* with 52.9% of correctly classified and 47.2% false negatives obtains worse results. A possible explanation to this is the low number of specimens for this species, and the variability in its presentation on the trays due to its greater than average length.

To better visualize the comparison between IS network configurations, Figs. 11 and 12 show per-species (per-class) AP score bars and F1 score, respectively. Bars in lighter blue shades represent 'classical' YOLACT, whereas bars in darker tones represent YOLACT++ configurations. It can be observed that the latter has a clear superiority in terms of AP scores for virtually all species. Furthermore, a larger backbone size with YOLACT++ seems to give it a minor boost.

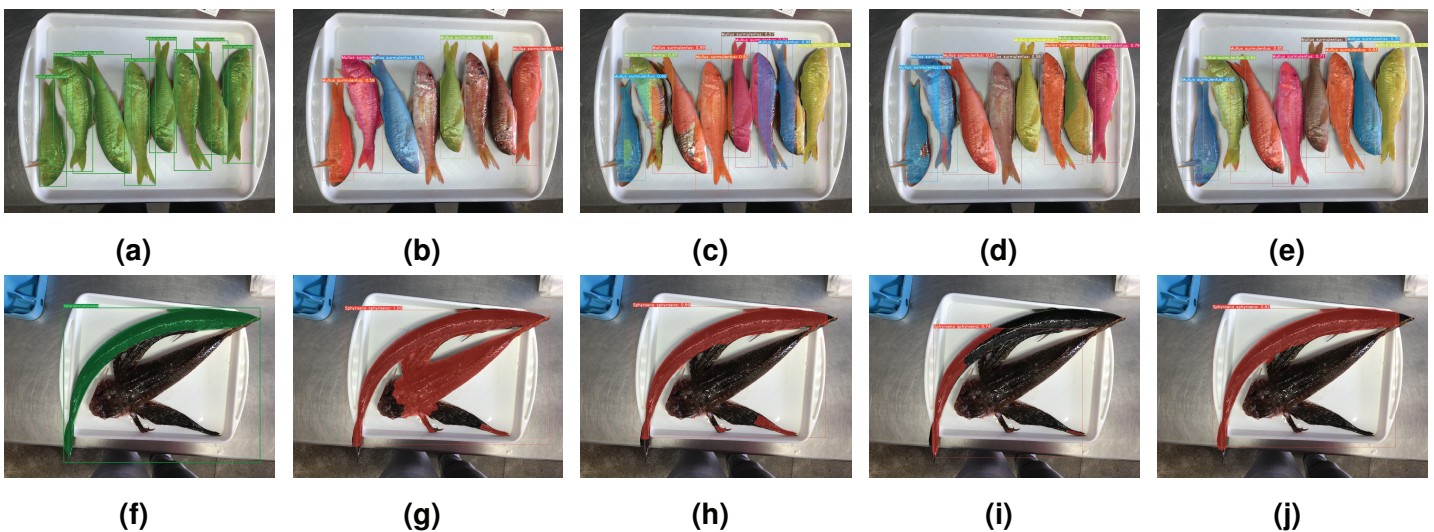

**Figure 13 Success and failure cases for segmented and classified specimens.** Successes, top row (A–E): ground truth (A); YOLACT ResNet-101 (B); YOLACT ResNet-152 (C); YOLACT++ ResNet-50 (D); YOLACT++ ResNet-101 (E). Failure cases, bottom row: (F–J): same order as top row. Best seen in colour.

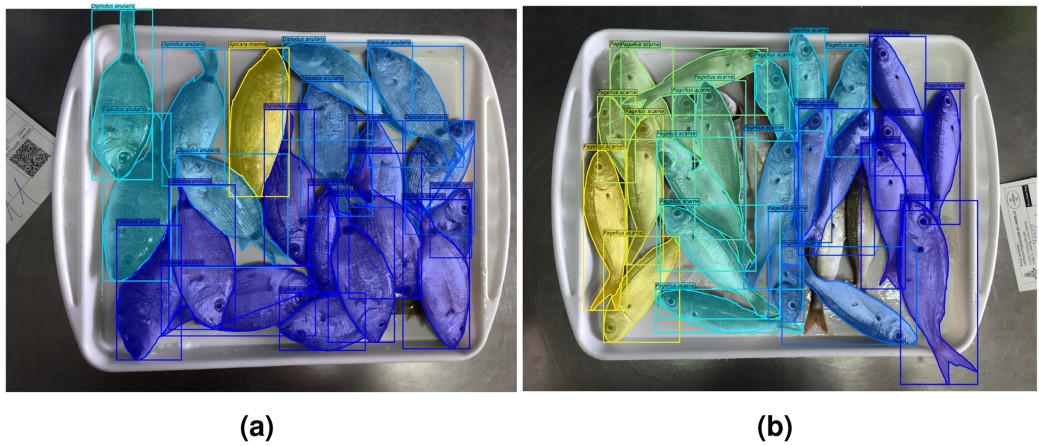

**Figure 14 Examples of fish trays with specimen overlap. Successfully labelled (A); and with some missing exemplars (B).**

Finally, for illustrative purposes, Figs. 13 and 14 show qualitative results. First, Fig. 13 depicts results for all tested network configurations on the IS module. The top row shows success cases with good segmentation and classification. Please note some fish in (b) are not fully detected with the simplest backbone used, but this is improved in (c), (d), and (e). The lower row shows examples of cases where the networks failed (possible cause is odd shape of *Sphyraena sphyraena*, combined with overlap). Furthermore, Fig. 14 shows images with overlapping specimens, and how this affects the behaviour of the IS module. On the left side, an example with good performance is shown, whereas the right image shows some missed detections due to heavy overlap.

**Table 2 Results for the fish size regression errors (in cm) of different regressors.** Best values are marked in bold.

| Model | MAE | MSE | $R^2$ | MAPE | Time (s) |
|---|---|---|---|---|---|
| Extra trees | 1.8613 | **8.7115** | **0.7694** | **0.1173** | 0.101 |
| CatBoost | **1.8506** | 8.8161 | 0.7668 | 0.1172 | 1.211 |
| Gradient boost | **1.8504** | 9.3102 | 0.7544 | 0.1166 | 0.075 |
| Random forest | 1.8830 | 9.5934 | 0.7474 | 0.1175 | 0.201 |
| Light GBM | 1.9224 | 9.5624 | 0.7471 | 0.1201 | 0.021 |
| XGBoost | 1.9853 | 9.8369 | 0.7409 | 0.1252 | 0.076 |
| $k$-NN | 2.0806 | 10.1672 | 0.7312 | 0.1331 | 0.005 |
| Linear | 2.5980 | 15.2516 | 0.6071 | 0.1656 | 0.127 |
| Ridge | 2.5973 | 15.2517 | 0.6071 | 0.1655 | 0.003 |
| Bayesian ridge | 2.5962 | 15.2524 | 0.6071 | 0.1655 | 0.003 |
| Least angle | 2.6365 | 15.518 | 0.5993 | 0.1676 | 0.003 |
| Huber | 2.4311 | 16.4486 | 0.5823 | 0.1585 | 0.005 |
| Decision tree | 2.6236 | 17.0694 | 0.5577 | 0.1617 | 0.007 |
| Lasso | 2.7333 | 19.2231 | 0.5162 | 0.1769 | 0.004 |
| Elastic net | 2.7526 | 19.3541 | 0.5145 | 0.1768 | 0.003 |
| OMP | 2.6763 | 21.8864 | 0.4456 | 0.1753 | 0.003 |
| AdaBoost | 4.3506 | 29.9264 | 0.1917 | 0.3018 | 0.035 |
| PA[a] | 3.8979 | 31.7804 | 0.1534 | 0.2338 | 0.004 |
| LLA[b] | 4.3817 | 39.4312 | −0.0028 | 0.2706 | 0.003 |
| Dummy | 4.3817 | 39.4312 | −0.0028 | 0.2706 | **0.002** |

**Notes:**
The best 20 of a total of 25 are shown, ordered by ascending mean square error (MSE). Error is provided using several common metrics (MAE, MSE, $R^2$, MAPE). The total time (Time) in seconds (s) is also provided for comparison of regression performance.
[a] Passive–aggressive.
[b] Lasso least angle.

## Regression results

As introduced in "Proposed Size Regression Experiments", five experiments were conducted. First, the performance of 25 regression models is analysed for the problem. The results are summarized in Table 2 which shows error rates for the best 20 models tested using a machine learning software package (*Scikit-learn, 2023*). Different common error rates with regard to the size are provided: mean absolute error (MAE), mean square error (MSE), the coefficient of determination ($R^2$), and the mean absolute percentage error (MAPE). Performance is also shown in terms of speed, by providing regression times in seconds (right-most column).

As a second experiment, the six best-performing regressors, and SVM (used as a baseline) will be fine-tuned to further improve the results from the previous experiment. These six regressors are: extra trees, gradient boosting, categorical gradient boosting (CatBoost), light gradient boosting (Light GBM), random forest, and extreme gradient boosting (XGBoost). The results for the selected regression models are shown in Table 3.

Next, the third experiment evaluates the selection of an appropriate normalization for the data. Three different normalizations have been tested: standard normalization (*i.e.*,

**Table 3 Comparison between results of the best six regression models considered (and SVM, as a baseline), when parameter tuning is applied.** Best values are marked in bold.

| Regressor | MAE (cm) | MSE | $R^2$ | MAPE |
|---|---|---|---|---|
| Extra trees | 1.8108 | 8.8154 | 0.769 | 0.1152 |
| GBR | 1.8339 | 8.7386 | 0.7705 | 0.116 |
| CatBoost | **1.8033** | **8.6005** | **0.7742** | **0.1144** |
| Light GBM | 1.8780 | 8.9229 | 0.7649 | 0.1188 |
| Random forest | 1.8329 | 9.0251 | 0.7645 | 0.1161 |
| XG Boost | 1.8452 | 8.8572 | 0.7662 | 0.1158 |
| SVM (baseline) | 1.9343 | 10.1436 | 0.736 | 0.1244 |

**Note:**
Best result in bold.

**Table 4 Comparative of MAE in centimetres between the best regression models analysed and different normalization of the data input and output.** Best values are marked in bold.

| Regression model | No scaling | Standard on input | MinMax on input | MinMax on I/O |
|---|---|---|---|---|
| GBR 10-fold | 1.8564 | 1.8564 | **1.8539** | 1.854 |
| Extra trees 10-fold | 2.0052 | 1.9969 | **1.9857** | 2.0119 |
| SVM 10-fold | 4.3581 | 1.8471 | **1.8195** | 21.5391 |
| CatBoost 10-fold | 1.7954 | 1.7920 | **1.7710** | 1.7824 |

subtraction of mean and division by standard deviation), as well as MinMax on the input, and MinMax on the input and output; which is performed by subtracting the minimum value and dividing by the range (max-min). Table 4 presents the results for this experiment, which show MinMax normalization on the input as the best-performing.

Contrary to other normalization schemes, MinMax does not change the shape of the distribution, preventing reduction in weight or importance of outlier instances in the model, which could explain its advantage in this case, given the particularities of some instances in the used dataset, which might be considered outliers, as per the common definition of this term, *i.e.*, errors in measurement or very uncommon instances. However, in the dataset used, some species like the above-mentioned *Sphyraena sphyraena*, which represents 2% of instances (124 fish, *i.e.*, can be considered rare), has annotated sizes that are generally larger than for all other species. Specimen lengths for this species are in the range of 25 to 83 cm ($\bar{x} = 45.00 \pm 12.62$ cm). Yet, in general, the dataset is in the 5 to 83 cm range ($\bar{x} = 17.00 \pm 6.91$ cm). As a consequence, all instances of *Sphyraena sphyraena* can be considered an *outlier*, as they are longer than most other fish.

In the fourth experiment, a 10 *k*-fold validation is applied on the MinMax normalized data from the previous phase. The results in Table 5 show the mean performance of 10 different 10-fold runs, with varying initialization seeds, to avoid possible situational errors due to causality (which explain the slight difference in the results). As in previous results, SVM is included as a baseline, but this time with two different kernels, linear and radial.

**Table 5 Final results with the best regression models analysed with the three original inputs (bounding box in pixels, segmentation mask area in pixels, species class label).** Best values are marked in bold.

| Regression model | MAE (cm) | $R^2$ |
|---|---|---|
| GBR 10-fold | $1.8501 \pm 3.0099$ | 0.7613 |
| Extra trees 10-fold | $1.9715 \pm 3.0396$ | 0.7462 |
| SVM linear 10-fold | $2.6711 \pm 4.4582$ | 0.4746 |
| SVM radial 10-fold | $1.8741 \pm 3.1885$ | 0.7307 |
| CatBoost 10-fold | $\mathbf{1.7614 \pm 2.7633}$ | **0.7926** |

**Table 6 Final results with the best regression models analysed with the three original inputs and calibration inputs, *i.e.*, four points of the tray ($x$, $y$).** Best values are marked in bold.

| Regression model | MAE (cm) | $R^2$ |
|---|---|---|
| GBR 10-fold | $1.3304 \pm 2.0937$ | 0.8740 |
| Extra trees 10-fold | $1.4098 \pm 2.3239$ | 0.8531 |
| SVM lineal 10-fold | $1.8234 \pm 3.3598$ | 0.6996 |
| SVM radial 10-fold | $1.2994 \pm 2.2449$ | 0.8620 |
| CatBoost 10-fold | $\mathbf{1.2713 \pm 2.0616}$ | **0.8840** |

Finally, in the fifth and last experiment, additional input fields are provided to the regressor. These inputs consist of data that would usually be unavailable, that is data regarding calibration, namely: coordinates of tray corners (or tray handle corners, more precisely). The idea behind the experiment is to assess how these four two-dimensional points can assist the regressor, and reduce error in the output bounding boxes and segmentation masks obtained. These errors are caused by the perspective, distance, and other image differences. The goal is to determine by how much do results improve when the regressor is provided with these data, even if they are part of the ground truth (*i.e.*, they were manually annotated), and cannot be therefore be automatically obtained by the system. Table 6 shows the results for this last experiment, and confirms that this information helps improve the results. This opens the interest for future automated tray corner detection systems. It also shows that the absolute error can be reduced by 0.49 cm when including this information, or conversely, that the uncalibrated system *only* performs 0.49 cm worse than the calibrated version. That is, depending on other constraints (*e.g.*, economical, time, *etc.*) it might be worth keeping an uncalibrated system, and sacrifice accuracy by a 0.49 cm margin.

## End-to-end results

Qualitative results from the whole system can be seen in Fig. 15, in which both outputs ($y_{NN}$ and $y_{size}$) are combined and visually represented. Furthermore, expert, statistical data from the field of marine biology is used in the form of size-to-weight charts to derive weight (biomass) of each fish instance, based on the regressed fish size.

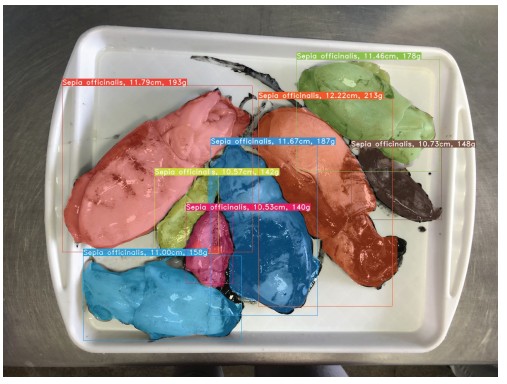 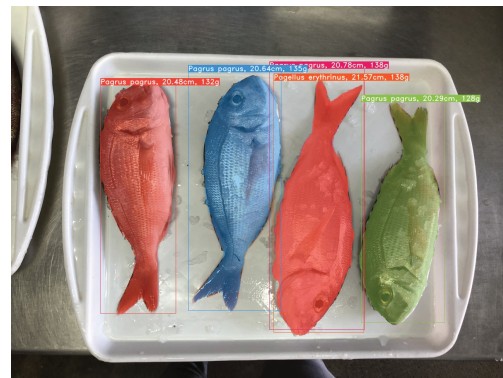

**(a)** Output for tray of Sepia off cinalis  **(b)** Output for tray of Pagrus pagrus

**Figure 15** **Example output images for the proposed system, in which masks, bounding boxes, species labels, and specimen sizes are shown for each detected fish instance.** Furthermore, using statistical data from the field, weight (biomass) is also provided, which is derived from size estimations.

## CONCLUSIONS

The main contribution of this article is the proposal of an end-to-end system for fish instance segmentation (IS), as well as fish size regression. The system relies completely on uncalibrated images at the time of inference for new images. To the best of our knowledge, there is no study performing automatic fish instance segmentation, species classification, and size regression from uncalibrated images for fish caught and presented in trays at fish markets. The results obtained so far are encouraging, and might be useful for a flourishing 4.0 fishing industry, which not only includes big players, but also small-scale, artisanal fish markets. Moreover, these techniques can generalize to other fields or scenarios, where IS and size regression are needed, and in which fitting and setting up fixed cameras is not possible.

To summarize the workflow, the system first efficiently uses the YOLACT family of neural networks, which has previously been trained from manually annotated data of fish species and correct fish instance segmentations. Additionally, visual metrology data is used to determine the homography, and therefore be able to convert pixel sizes to real-world sizes in centimetres during training of fish size regression. The data thus collected from the neural network and the visual metrology are then used to train the regressor. During the inference of new images, uncalibrated images are used, and all information, *i.e.*, fish species labels, instance segmentation, and fish sizes are obtained.

The proposed method avoids the use of visual metrology during inference, which would require a fixed calibrated camera, or the use of corner markers in fish trays or other visible 'token' objects in the image, for on-the-fly calibration of the image.

This lack of calibration at inference is justified by the nature of some wholesale fish markets, especially smaller ones, since artisanal markets lack the infrastructure (*e.g.*, conveyor belts, digitized auctioning systems, *etc.*). The ultimate goal, here, is to foster the digitization of traditional and artisanal fishing industries, and provide them with reliable

and thorough data on fish catches, sales, weights, *etc*. Therefore, the method proposed here represents a first step towards this more ambitious series of systems for the digital management of fisheries. For validation of the proposed method, the DeepFish dataset is used, which includes a large amount of annotated images of fish trays from a local fish market. This is publicly available, and provided to the community for further research into other similar problems, as well as for other more general applications.

Using this annotated data, and data derived from it, the IS and regressor modules have been trained. The point of the IS evaluation was to show the performance of a set of YOLACT variants, and demonstrate their utility for the task at hand, and see the impact of different backbones in the performance and inference time. Results show that, the best performance were obtained using YOLACT++, with the larger ResNet-101 backbone, which discards the hypothesis of the larger backbone. Furthermore, results also show that it is possible to detect interspecies subtleties *e.g.*, fishes of the Mullus genus, *i.e.*, *M. barbatus* and *M. surmuletus*, are very similar, but correctly identified with high confidence; or *S. tinca* specimens being correctly distinguished by sex. In general, results are very promising with the proposed solution, in terms of instance segmentation, and species classification.

With regard to the fish size regression, categorical gradient boosting regression (CatBoost) has been proven to be the most suitable model for the problem, after normalization (using MinMax), and hyperparameter tuning. Furthermore, additional experiments have shown that regression results improve when additional real-world fish tray size data (*e.g.*, handle corner points, or similar) are included as additional inputs to the regressor, since the IS module works on uncalibrated images of similar characteristics. These experiments show that this data, albeit unavailable at inference time in our system, might be useful to better tune the fish regression module, as it contains valuable information regarding the real-world sizes. This opens lines for future work in this regard.

One line for future work would be to include an automated tray corner location module. However, when fitting the system in larger-size fish markets in the future, it might not be necessary to have this module, as it may be possible to use fixed cameras over pre-existing facilities such as auction conveyor belts. It would therefore be an optional module in the system. Other lines of future work include, in the short term, calculating biomass extraction rates (total, and per-species) based on estimated fish sizes, or similarly *via* areas (from masks) or volumes (if using depth information). Furthermore, in the medium term, geographical vessel information, related to fish batches, is to be included in the analyses to better understand the availability and status of fishing stocks in a certain area.

### Funding

This work was developed with the collaboration of the Biodiversity Foundation (Spanish Ministry for the Ecological Transition and the Demographic Challenge), through the Pleamar Programme, co-financed by the European Maritime and Fisheries Fund (EMFF) Deepfish/Deepfish 2 projects. The European Regional Development Fund (ERDF) and

MCIN/AEI/10.13039/501100011033 supported this research under the "CHAN-TWIN" project (grant TED2021-130890B-C21) and the HORIZON-MSCA-2021-SE-0 action number: 101086387, REMARKABLE, Rural Environmental Monitoring *via* ultra wide-ARea networKs And distriButed federated Learning. The funders had no role in study design, data collection and analysis, decision to publish, or preparation of the manuscript.

### Grant Disclosures
The following grant information was disclosed by the authors:
Biodiversity Foundation (Spanish Ministry for the Ecological Transition and the Demographic Challenge).
Pleamar Programme.
European Maritime and Fisheries Fund (EMFF) Deepfish/Deepfish 2 Projects.
The European Regional Development Fund (ERDF): MCIN/AEI/10.13039/501100011033.
"CHAN-TWIN" Project: TED2021-130890B-C21.
HORIZON-MSCA-2021-SE-0: 101086387.
REMARKABLE, Rural Environmental Monitoring *via* ultra wide-ARea networKs And distriButed federated Learning.

### Competing Interests
The authors declare that they have no competing interests.

### Author Contributions
- Pau Climent-Perez analyzed the data, prepared figures and/or tables, authored or reviewed drafts of the article, and approved the final draft.
- Alejandro Galán-Cuenca conceived and designed the experiments, performed the experiments, performed the computation work, authored or reviewed drafts of the article, and approved the final draft.
- Nahuel E. Garcia-d'Urso performed the experiments, performed the computation work, authored or reviewed drafts of the article, and approved the final draft.
- Marcelo Saval-Calvo analyzed the data, prepared figures and/or tables, authored or reviewed drafts of the article, and approved the final draft.
- Jorge Azorin-Lopez conceived and designed the experiments, analyzed the data, prepared figures and/or tables, and approved the final draft.
- Andres Fuster-Guillo conceived and designed the experiments, analyzed the data, prepared figures and/or tables, and approved the final draft.

### Data Availability
The DeepFish Dataset is available at Zenodo: Andrés Fuster-Guilló, Jorge Azorin Lopez, Nahuel Emiliano D'Urso, Alejandro Galan Cuenca, Gabriel Soler Capdepon, Maria Vicedo Maestre, Juan Eduardo Guillen Nieto, & Paula Perez Sanchez. (2022). DeepFish Dataset (April 2022 update) (v3.1) [Data set]. Zenodo. https://doi.org/10.5281/zenodo.6475675.
The code is available at Zenodo: García-d'Urso, N. E., Galán-Cuenca, A., Climent-Pérez, P., Saval-Calvo, M., Azorin-Lopez, J., & Fuster-Guillo, A. (2023). DeepFish project Yolact-

based fish instance segmentation and species classification. Zenodo. https://doi.org/10.5281/zenodo.10102149.

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
