# Peer review of "Simultaneous, vision-based fish instance segmentation, species classification and size regression"

_PeerJ Computer Science, doi:10.7717/peerj-cs.1770_

## Round 0.1 · original submission · Major Revisions

Based on the referee reports, I recommend a major revision of the manuscript. The author should improve the manuscript, taking carefully into account the comments of the reviewers in the reports, and resubmit the paper.

**Language Note:** The review process has identified that the English language must be improved. PeerJ can provide language editing services - please contact us at copyediting@peerj.com for pricing (be sure to provide your manuscript number and title). Alternatively, you should make your own arrangements to improve the language quality and provide details in your response letter. – PeerJ Staff

Reviewer 1 ·

Basic reporting

Summary

This paper describes a system for automatic fish instance segmentation and size regression in pictures taken at a fish market. The system is tested against different implementation options for the instance segmentation network and different types of regressors. A dataset of fish pictures with manual annotations has been collected for the experiments. The obtained results illustrate the performance of the system.

Review

The paper is well written in general. The proposed system is described with clarity and the experiments are correct. However, a few issues, listed below, must be addressed prior to publication.

Main comments:

1) Some important details are missing in the description of the Method and the Experiments:

1.1) You mention that input images are pre-processed, both at the edge layer of the system (l.165) and before training (l.187). You should give details about this pre-processing step.

1.2) In l.216 you say that rotation and translation parameters for the fish trays need to be estimated. However, since you want to estimate an homography you also need to compute scale and shear parameters.

1.3) In l.240 it is mentioned that a RGB-D camera is used to capture the images. However, you don't use depth information in this paper. This should be clearly stated, since the use of this type of camera may confuse the readers.

1.4) In l.281 you state that "Since the dataset is highly imbalanced (...) data augmentation is used to train the neural network". This means that after performing the described augmentations the dataset is balanced (e.g. the classes with less fish undergo more augmentation transforms that the classes with more fish, so in the end the number of fish in each class is the same)?

1.5) The formula for the binary cross entropy (Eq. 1) should be included in the text

1.6) The regression module should be described in more detail:
- you claim that the input of this module is "consists of the masks, bounding boxes, and species labels". The output is the estimated fish size in cm? But, how can you distinguish the cases where two bounding boxes contain fish of the same size, but one belongs to an image taken at a farther distance of the tray than the other? The bounding boxes will have different sizes. Without information about the rest of the image (or the size of the trays, as in one of your experiments), the reliability of the results depend on taking all the pictures roughly at the same distance from the trays and with the cameras similarly oriented.
- you talk about three different types of normalization of the data in the regression module. How are these normalizations performed: standard is substraction of mean and division by standard deviation?, minmax mean linear transform the minimum to 0 and the maximum to 1?
- can you give references for all the methods listed in Table 2? Have you used a software package where these methods are implemented?

1.7) In the Experiments, can you include some examples when the fishes overlap, and comment the behaviour of the system?


2) In the Previous Work section, you should mention that the Mask R-CNN network has also been used for instance segmentation of fish (e.g. Alvarez-Ellacuria et a. (2020), and French et al. (2019))

Other comments:

1) English writing. A careful revision is needed, since the paper contains several typos and grammatical mistakes. Some examples: "are deemed" should be "is deemed" (l.70), revieww (l.84), "my" should be "may" (l.235), "the system need" should be "the system needs" (l.244), "by low" should be "by the low" (l.388),
"larger that all other" should be "larger than for all other" (l.435),

Moreover, I don't understand this sentence (l.210.211): "these become a new X, that is an X of samples ..."

2) When refering to the different sections of the paper, the identifiers of the sections are missing or wrong: references to Sec. 0.0.1 in lines 157, 162, 226, 280, 414; missing label in l.168.

Experimental design

No additional comment.

Validity of the findings

No additional comment.

Reviewer 2 ·

Basic reporting

no comment

Experimental design

no comments

Validity of the findings

no comments

Additional comments

I have attached a pdf with my comments

Annotated reviews are not available for download in order to protect the identity of reviewers who chose to remain anonymous.

·

Basic reporting

On the Narrative and Focus: I advise the authors to identify a more precise focus for the paper. Currently, it seems to be caught between different categories. Clarifying the main aim and target audience would help in creating a more cohesive narrative. Is the target audience fisheries ? if so drop the model comparison and focus more on the application. Is the focus computer vision research then focus on setting the aim to do a systematic review of different models.

Writing Quality: I recommend engaging a scientific proofreader or editor to improve the clarity and precision of the writing. The writing is often imprecise or too casual.

Literature Review: I recommend that the authors expand the literature review to situate the work within the context of existing research. A more comprehensive literature review is required.

Experimental design

Regarding Model Validation and Testing: I strongly recommend incorporating standard validation techniques like cross-validation, out of sample testing or hold-out validation to prevent potential overfitting. Without validation or testing on unseen data, it is difficult to gauge the generalisability of the model.

About Size Regression: I see potential novelty in the size regression part of this work. I suggest that the authors expand on why this is novel and provide more detailed explanations to showcase how it can be applied more broadly.

Concerning Generalisability: Please add a section to discuss the potential limitations of the model and its performance in different or unconstrained environments. This discussion is vital to understanding the full scope of the model's applicability.

Validity of the findings

Overall Presentation: I believe there is a valuable method worth publishing in this paper, particularly with regard to length estimation, but the current presentation needs refinement. I recommend a major revision and provide specific guidelines above to improve the paper to a publishable standard below in Additional comments section 4.

There is no mention of model validation or testing. The results
seem to be as if they have been produced from the training data. If
this is the case, the model is very likely to be over fit. It is
standard practice in data science to make some effort to quantify
the generalisability. data-driven models are prone to over fitting
and unless the authors address this, I find it hard to justify
publishing.

Additional comments

1 General
=========

- There is a mixture of Fig. and Figure throughout the text


2 Abstract
==========


3 Introduction
==============

L == Line number

L 47 - 48: This is a bit vague. What are the specific problems L 49:
How does a computer help reduce errors. An inaccurate model will
increase errors.


4 Previous work
===============

L 59: 'A large amount' is not scientific language. use comparative
language such as 'larger than ...' or quantitative language. what is
considered 'large' ? L 61: If considering image based size
estimation, consider the following citation
<https://doi.org/10.3389/fmars.2023.1171625> L 69: Who is the audience
here, if the abstract discusses challenges around fish stock
assessment, then instance segmentation is the focus of the problem its
a tool potentially used to address the problem of size estimation L
84: typo 'revieww' L 90: Check this reference. this refers to
YOLO9000, I believe YOLOv3 (earlier) was Redmon J., Divvala S.,
Girshick R., Farhadi A. (2016). “You only look once: unified,
real-time object detection,” in Proceedings of the IEEE conference on
computer vision and pattern recognition. 779–788 l 90: if using YOLO
for species recognition, consider the following citation
<https://doi.org/10.3389/fmars.2022.944582> In fact, this entire
section is very lite on reference with regarding to use ML for species
detection. Take a look at section 1.2 of that paper for more
references l 95: 'good results' too casual language. See comments
above. A lot of language used in the manuscript is too causal or
qualitative This section needs to end by tying it back to the problem
they are trying to solve around fish stock assessment, and why these
previous studies haven't yet solved it and why this manuscript either
builds on or improves. There needs to be more context in general with
this section and perhaps more critical of the literature.


5 Proposal
==========

L 105 - 106: Is it ? why do we need to automate fish instance
segmentation. The story needs a bit of work here. Set up the problem
more clearly. We really need to measure the size of fish. Instance
segmentation MAY help with this, but the authors have yet to convince
the reader how or why. L 106: Same with cloud and edge computing.
Why is the reader reading about 'end nodes' ? L 112: performance of
what ? speed, accuracy, latency ? etc L 114: move this to the
beginning of the section. L 115: 'Section ' missing section number
and cross reference. L 116: instance segmentation doesn't include
species classification. Your application of the model does both. This
sentence needs rewording. L 116: What is 'fish size regression' and
how does Figure 1 help explain that. Again who is the audience here, a
systems architect or someone interested in fish size from computer
vision ? L 122: instance segmentation has already been used so put
the 'IS' in the first use of the term L 134: missing space at the end
of the sentence L 134: remove 'so' L 156: section (Sec. ) ? Missing
section number ? L 157: Why start at 0.0.1. Consider putting numbers
for each major section and then use (Section, Sub-section,
sub-sub-section) numbering L 161-162: 'will be presented' tense seems
wrong here. the results are presented in this manuscript.


6 Method
========

L 182: This should be in the literature review.


7 Dataset
=========

L 264: '100 specimens per species is considered necessary to train the
neural network.' Where did this number come from? I believe Alexy
(Yolov3 maintainer) recommends minimum of 1500 images per class. 100
is far too low. Figure 5. This is a very low number of images per
species. A common challenge, but I am sceptical of any kind of
genralisability from such a small dataset, even with Data augmentation
which is standard practice in many frameworks (Yolov5 does this by
default)


8 Augmentation
==============

L 280: why start numbering now and why start at 0.0.1


9 Proposed IS experiments
=========================

I am not sure what the purpose of this section is. If the aim of the
paper was to do a comparative study of different instance
segmentation models, then fine, but the context and tone of the paper
changes. If the aim of the study is to develop a model of species
classification and size then its sufficient to say a number of models
were assessed and present the details of the one that performed the
best. As scientists we understand we need to experiment with models
but we log those in our laboratory book.


10 Proposed size regression experiments.
========================================

L 328 - 334: this should be in the literature review. Justification
for their inclusion of this study should be made and then present how
they are used in the method in this section.
- This is potentially the most interesting part of the study but there
details are not described here in sufficient detail.


11 Results and Discussion.
==========================


12 IS results
=============

- Testing different backbones of Yolocat is a new experiment
- L 351: Language here is a bit too casual.
- L 354: lower case 'f' on figures
- Figure 6. Is this the mAP of the training data ? where is the
validation and test data results. How have these been defined.
- Figure 11. As before, are these the results of mAP of the full
corpus of data. It's standard practice to split into train,
validation and test data sets. Better yet, in and out of sample
testing should be conducted.
- Why are mAP scores reported and not F1 score for species
inference. the F1 score helps our understanding of how well the
model is calibrated and balanced between precision and recall.


13 Regression results
=====================

- using R^2 as from an ensemble of different models is not a robust
way to validate the results there cold be a bias in the data. Has
the data been validated again a non computer vision method like
manually measuring the fish with a tape measure. This would be the
simplest and most robust. If that was not possible then it needs to
be addressed.
- This was perhaps the most interesting part of the study. Being able
to measure fish from single, uncalibrated cameras has a very real
use-case. But without validation I can't see it being applied.
Perhaps a missed opportunity to make this the focus of the study.
- L 436: mixed precision here. average x value rounded to whole
number but the uncertainty is given to two decimal places. $\bar{x}
= 45.00 \pm 12.62$ for example. Table 5 does this correctly.


14 Conclusions
==============

- Without a more robust validation of both the species inference and
the uncalibrated length measurements most of the conclusions are
drawing a long bow. I find it difficult to see an end-to-end
solution here. That kind of solution would require a camera and
hardware solution, algorithm (validated) and software to with
support to users of the system.


14.1 Other Notes
~~~~~~~~~~

- There is no mention of model validation or testing. The results
seem to be as if they have been produced from the training data. If
this is the case, the model is very likely to be over fit. It is
standard practice in data science to make some effort to quantify
the generalisability. data-drive models are prone to over fitting
and unless the authors address this, I find it hard to justify
publishing.
- This paper hasn't quite got it's narrative right. It's not quite a
pure AI/ML focused paper which may conduct a systematic review of
instance segmentation models or introduce a new novel model.
Neither is it quite a methods paper which aims to succinctly describe
an application of computer vision to fish ecology. It's somewhere
in between.
- The size regression has the potential to be the most novel part of
the paper.
- It is not well written and would benefit from a scientific proof
reader.
- No discussion about model generalisaability, out of sample testing
application to unconstrained environments.
- How have they validated their results for size ?
- Why segmentation for size ?
- The language is a little imprecise at times.
- very light on literature review
- I think there is a method worth publishing here, but the story isn't
well presented.

---

## Round 0.2 · accepted · Accept

The author has addressed the reviewers' comments properly. Thus I recommend publication of the manuscript.

Reviewer 1 ·

Basic reporting

The authors have addressed all of my concerns in the new version of the manuscript.

Experimental design

The authors have addressed all of my concerns in the new version of the manuscript.

Validity of the findings

The authors have addressed all of my concerns in the new version of the manuscript.

Additional comments

The authors have addressed all of my concerns in the new version of the manuscript.

Reviewer 2 ·

Basic reporting

The authors have incorporated all the modifications recommended by the reviewers

Experimental design

The authors have incorporated all the modifications recommended by the reviewers

Validity of the findings

The authors have incorporated all the modifications recommended by the reviewers